# Well Played! Promoting Phonemic Awareness Training Using EdTech—GraphoGame Brazil—During the COVID-19 Pandemic

**DOI:** 10.3390/brainsci12111494

**Published:** 2022-11-03

**Authors:** Juliana G. Marques de Souza, Janaina Weissheimer, Augusto Buchweitz

**Affiliations:** 1Department of Modern Foreign Languages, Federal University of Rio Grande do Norte, Natal 59078-970, Brazil; 2Brain Institute (ICe), Federal University of Rio Grande do Norte, Natal 59078-970, Brazil; 3Department of Psychology, University of Connecticut, Stamford, CT 06901, USA; 4Haskins Laboratories, New Haven, CT 06511, USA; 5BraIns (Brain Institute of Rio Grande do Sul), Federal University of Rio Grande do Sul, Porto Alegre 90610-000, Brazil

**Keywords:** phonemic awareness, early literacy, GraphoGame, school closure, EdTech

## Abstract

Early literacy skills such as alphabet knowledge and phonemic awareness are made up the foundation for learning to read. These skills are more effectively taught with explicit instruction starting inpreschool and then continuing during early elementary school years. The COVID19 pandemic school closures severely impacted early literacy development worldwide. Brazil had one of the longest school closure periods, which resulted in several children having no access to any educational activities. Education Technology (EdTech) tools can leverage access to pedagogical materials and remediate the consequences of school closure. We investigated the impact of using an early literacy EdTech, GraphoGame Brazil, to foster learning of early literacy skills during the height of COVID19 school closures, in Brazil. We carried out a quasi-experimental, pretest and posttest study with elementary school students who were taking online classes. Participants were pseudo randomly assigned to (1) an experimental group, who played GraphoGame Brazil, and to (2) an active control group, who played an EdTech that focuses on early numeracy skills. The results show a significant positive training effect on word reading accuracy associated with the use of GraphoGame for the children in the experimental group, relative to the control group. We also found statistically significant negative effect in lowercase naming for the control group. We address the consequences of COVID19 school closures, the promise of EdTech and its limitations, and discuss the issue of fostering successful early literacy instruction in countries that have struggled with teaching children to read even before the pandemic.

## 1. Introduction

The early steps of teaching children to read require explicit instruction of specific skills, which in their turn are the foundation for breaking the written code. Learning to read is different from the acquisition of oral language, the latter is a natural skill whose development is supported by “plasticity expectant” structures in the brain [1]; in other words, oral language development relies on hardwired human brain structures that draw on naturally occurring oral language stimulation [2]. Oral language development does not require explicit instruction. It is formidable in its reliance on human interaction alone. However, learning to read requires explicit instruction. Literacy skills do not develop by exposure alone to books and the world of print [3].

Brain imaging studies have consistently shown that though reading is not hardwired in the brain, on the “reading brain” largely overlaps with the brain’s networks for oral language.

The overlap between oral language and reading-related brain networks has been shown in several languages, independent of the writing system; moreover, the more proficient the reader, the more the two systems overlap [1,4,5,6,7]. Just as the instruction-dependent process of learning to read draws on foundational oral language skills, such as phonological awareness, the adaptation of a child’s brain to reading “piggybacks” on the existing language networks [4,6,8,9,10]. Moreover, delays in oral language acquisition are associated with poorer outcomes in reading and writing skills. They also represent early signs of risk for developmental dyslexia [11,12,13,14,15]. Explicitly teaching children to bridge oral language sounds with the more fine-grained phonemes is a critical pedagogical goal for effective early literacy development. It is at the heart of successfully teaching more children to read [16,17,18,19,20,21].

One of the key processes of adaptation of the brain to reading is in finding a “home” in the brain for the identification of strings of letters. When infants learn to read and their brains encounter visual stimuli, “a subset of visual regions becomes specialized in recognizing strings of letters and sends them to spoken language areas” [3]. What is special about strings of letters in comparison to other visual objects? 

The plasticity of portions of the occipital and temporal lobes system of the brain leverages one adaptation that is crucial to reading, i.e., breaking our mirror invariance for visual objects [5,11,12]. Pre-reading children are, of course, able to identify and name objects and beings. However, naming letters requires that children break with the human brain’s natural mirror invariance for visual objects [22,23,24]. The human brain is hardwired to allow for recognizing mirror images. We see ourselves in the mirror and have no trouble recognizing our faces: Yet, “b” and “d” are mirror images of one another but children need to tell them apart. As children learn to read, the brain needs to be able to set “b” and “d” apart and identify that, say, if “E” is written with the horizontal lines to the left of the vertical line, it is no longer “E” (it is a trident, maybe?). The misleadingly trivial process of breaking mirror invariance does not develop naturally, in the absence of instruction. When children begin to learn about letters, there is a portion of the brain’s occipitotemporal area that adapts to the specific process of identifying the mirror-invariant letters and letter strings. The occipitotemporal region that adapts to word form recognition and concentrates learned knowledge of letter strings is also known as the Visual Word Form Area (VWFA) [11,13,14,15,16,17,18]. Letter naming and sound identification is key for the early stages of GraphoGame, in which single-letter sound recognition and uppercase and lowercase letter identification is practiced. As the game progresses, so does the challenge that draws on phonemic awareness and, ultimately, word-level decoding.

Like letter identification and naming, phonemic awareness is another key skill for preparing children to read [2]. It is the result of the invention of the writing system, which relies on a more fine-grained representation of language sounds than the, say, more coarse phonological awareness. The development of phonemic awareness requires explicit instruction focused on identification and manipulation of grapheme-phoneme representations. Phonemes are the units of sound that humans have represented with letters; again, they are the result of the invention of script [21]. Skillful phonemic awareness involves being able to distinguish and manipulate these phonemes [18]. In GraphoGame, several levels named “Word Forming” practice the identification of letters that make up syllables and words.

### The Promise of Education Technology and COVID19 School Closures

Twenty-first century children are born into a world of technology. Other than children who are underserved by access to technology, most have experienced living in an environment of computers, smartphones, tablets and videogames. In this sense, Education Technology (EdTech) has been a hot topic for innovation in education. However, the use of EdTech is not always supported by reliable research. GraphoGame is supported by extensive research [25,26,27,28], and it is effective for early literacy instruction, especially when used by the teacher or in interaction with a tutor or adult [29].

The recent COVID-19 pandemic has affected education in several levels, especially among children of lower and lower-middle income countries. School closure in Brazil was among the longest. To be sure, children who were underserved by access to technology before the pandemic were also the most affected by school closures. In Brazil, a survey showed that in 2020, 13.2% of children aged 6 to 10 years were out of school or had no school activities; the figures were higher in the more poverty-stricken regions of the North (26.9%) and Northeast (16.1%) [30]. Moreover, the percentage of children aged 6 and 7 years who did not learn to read increased from 25.1% in 2019, to a 40.8% in 2021 [31]. Students in institutions that rapidly transitioned from face to face to web learning were the least affected by consequences of school closure [26]. In Brazil and other lower-middle- and lower-income countries, the access to online education and EdTech is intertwined with socioeconomic status; for example, 51% of 6–7-year-old children who are not reading are from the lowest income quartile, whereas 16.6% are from the highest income quartile [31]. Moreover, school closures disproportionately impacted poorer, rural area children in Brazil [30]. 

GraphoGame (GG) is an EdTech developed by researchers at the University of Jyväskylä, and has been shown to support early literacy instruction in different cultural contexts. GG emerged from the Jyväskylä Longitudinal Study of Dyslexia to assist children with the challenge of learning to read, at first, especially children at risk for developmental dyslexia. With time, GG showed it could be used to foster early literacy instruction in general. It is currently delivered as an APP in several languages and operational systems [29]. The languages include English (American and British), French, Spanish (in Chile, Argentina, among other Spanish speaking countries), European Portuguese, Norwegian, Polish, Pinyin, among others [25,27,28,32,33].

We used GraphoGame Brazil, which is the Brazilian Portuguese version of the EdTech. The version used in the study was the 2020 version 1.2 which consisted of over 40 streams. Its logic, like other GG versions, involves incremental challenges and reinforcements. GG Brazil starts with uppercase vowel and consonant single letter levels, then introduces lower-case vowels and consonants, simple and complex syllables; subsequently, it introduces words, also in increasing levels of difficulty (note: the most recent version includes images of mouth articulation, phrases and sentences, which were not in the 2022 version). Of note, GG Brazil originally introduces uppercase and lowercase identification. A more recent version (released after this study) also provides mouth-articulation images to help differentiate letter sounds (like “m” and “n”) and practices phrases and full sentences. In Brazil, children are taught to read using uppercase letters only. Lowercase letters are not introduced until the 3rd grade. This educational practice is not supported by any pedagogical or psychological evidence; it stems from anecdotal understanding that introducing uppercase and lowercase at the same time “confuses” children. As of the submission of this study, the APP was the most downloaded version of GG in the world with over 1.5 million downloads. 

We carried out a quasi-experimental design pilot study to understand the relationship between using GraphoGame and the development of early literacy skills and decoding words in Brazilian Portuguese, during school closures and the height of the COVID-19 pandemic in the country, i.e., from February to May of 2021. We hypothesized that GG would positively impact children’s early literacy skills, as measured by the tests in the study and, especially, due to the challenges of school closure and teaching children online; likewise, we expected that the total time played would be associated with the gains in literacy skills.

## 2. Materials and Methods

### 2.1. Study Design

There were two main research questions: (1) does GraphoGame positively impact training of phonemic and phonological awareness, letter recognition and word reading more than just traditionally attending classes? (2) Is there an association between total hours played and gains in early literacy skills and word reading?

We carried out a quasi- experimental study and the hypotheses were: (1) training with GG will positively and significantly impact children’s phonemic and phonological awareness, letter recognition and word reading more so than business as usual; (2) Total training time (total duration of play) will positively correlate with gains in phonemic and phonological awareness, letter recognition and word reading.

The protocol involved a 6-week training intervention with GraphoGame Brazil. Children participated in five 15 to 20-min sessions for the 5 days of the school week. Participants played their assigned games at home and were supervised by their guardians. We evaluated phonological and phonemic awareness using a standardized test in Brazilian Portuguese (CONFIAS) [34]. We also evaluated word reading (TDE2) [35] and Rapid Naming (NAR) [36]. 

We also applied a teacher-based screener for reading difficulties and risk for dyslexia [36]. The screener is based on teacher assessments of students reading and writing development. Total time played stats were collected from the statistics provided by the GG Brazil research license. The research license has the exact same content as the freely available GG Brazil, but it adds a layer of user and administrative access provides statistics on hours of play, accuracy, and game levels played, for instance. We only used GG research licenses with children whose parents or guardians provided a signed informed consent form for participation in the study. 

### 2.2. Participants

Our study started with a convenience sample of 60 children from 1st to 2nd grades. The study began in February, which in Brazil is the beginning of the school calendar. Participants who either withdrew during (incomplete pretests), or after (no posttests) the study or were excluded for not meeting minimum duration of play of 300 h, or adherence to the study (playing every weekday). Ultimately, 34 children met participation criteria in the study.

### 2.3. Procedure 

Due to school closure, the interaction with guardians was only possible online. We arranged virtual meetings to explain the research and answer questions. We had a pre-pandemic consent form which was we redesigned to an online format sent to parents or guardians. We also developed a verbal assent protocol to explain the experiment to the participants, who were asked to provide their verbal assent to participate in the study in addition to their guardian’s written informed consent. At first, 93 families showed interest in participating in the study by signing the online consent form given to them during the online meeting.

Both pre and posttest were preferentially conducted in-person in the schools. If necessary, we collected data through videoconferencing. We followed the health and safety protocol of the school for in-person data collection. However, all students played GG at home. Despite our efforts, only sixty children completed all the pretests and were subsequently pseudo randomly assigned to the experimental and active control groups (*n* = 30 each group). We excluded 33 participants who withdrew voluntarily during pre-testing or did not attend pretests sessions virtually or physically. 

The intervention group (IG) was assigned to play GraphoGame Brazil in daily sessions for 6 weeks, at home. In-person classes were irregularly scheduled at the time, and we could not rely on students attending or even the school staying opened for the duration of the study. The GG or control sessions were carried out after regular online school hours. Each session lasted 15 to 20-min and students were assisted by parents or guardians while they played. Parents and guardians were instructed by the researcher on how to supervise their children during the sessions. The instructions involved showing how to start the game, click on their corresponding avatar and insert a 3-digit pin-code, which ensured the player always started from the last played level, and that the data was uploaded to the server. Parents were explicitly instructed to assist children while playing, but also that they should not help participants if they made mistakes or were struggling in a level. Of note, GG detects irregular playing behavior. It flags the data and provides a warning to users to avoid sharing the avatar. 

We were available, online, to guardians and participants the entire duration of the session. We also provided daily reminders that the session was about to start. Children used earphones and were instructed to play in a quiet room. The active control group received the same instructions and assistance. They played the online numeracy development game called Vektor™ (at the time of the study, downloaded from cognitionmatters.org). Participants in the active control group also played 15 to 20 min sessions for 6 weeks. We provide an illustration of the study design in Figure 1.

We pseudo-randomly controlled the distribution of participants to balance for students in 1st and 2nd grades. All participants took the same pre and post intervention tests. Following the 6 weeks of intervention, some participants did not meet the core criteria of minimum play (300 h) and adherence (play every weekday). We were not able to reach three participants to conduct the post intervention tests. Participants who did not meet the playing time and adherence criteria were excluded. The final sample included thus 16 participants in the experimental group and 18 in the active control group. Compliance, thus, impacted our final sample to a loss of nearly 50%. It is challenging to assess whether the compliance was low or high. Compliance has been shown to vary according to groups, duration of study, among other factors, from 80% to a very low adherence of 16% at the end of year-long studies; compliance in nonclinical children and youth studies is of approximately 78% [37,38]. 

### 2.4. Instruments 

We carried out the following pretests: (1) participants read a list of words aloud for assessment of word reading accuracy and speed (standardized test called *Teste de Desempenho Escolar* (TDE II [35]); (2) participants named letters from the rapid naming test (*Nomeação Automática Rápida*—NAR [36]); and (3) participants took a standardized phonemic and phonological awareness test in Brazilian Portuguese (*Consciência Fonológica: Instrumento de Avaliação Sequencial* (CONFIAS [34]).

#### 2.4.1. TDE II Tests [35]

Children were asked to read the words aloud as fast as they could. If the participants had any uncertainties about how to read the words, they should read each one in the way they thought the word should be pronounced. We carried out a practice test to ensure participants understood. We ended the test if the student made 10 subsequent reading errors. Accuracy was measured based on the number of correct words read by the participants, out of 36. The session was recorded for subsequent review. 

#### 2.4.2. Letter Naming [36]

The letter naming test (NAR) assesses letter recognition. It has 50 items that are presented to participants, who are instructed to name each letter to their best of their knowledge. Participants were given a practice run. We evaluated efficiency (correct letters per second), and overall accuracy (number of correct letters recognized).

#### 2.4.3. CONFIAS: Phonological and Phonemic Awareness [34]

We assessed phonemic and phonological awareness with a standardized test for Brazilian Portuguese. Basically, participants were shown images, which they named, and subsequently tested for identification of phoneme similarity (e.g., first phoneme) from a set of pictures. The tests had 4 different stimuli and students are graded from 0 to 16 points. Children’s responses were recorded, and test scores were noted. 

#### 2.4.4. Total Time Played

The number of hours played by each child during the 6-week intervention with GraphoGame was extracted from the game stats to assess total playing time. After the 6-week intervention period, we repeated all tests. 

#### 2.4.5. Screener for Reading Difficulties [39]

We asked teachers to provide their assessment of the students using a screener for reading difficulties and dyslexia developed in Brazil [39]. We created an online version of the questionnaire which was e submitted by the teachers via Google forms. We used the answers provided by them to gather addition information regarding students reading abilities.

### 2.5. Data Analyses 

We analyzed test scores for normal distribution. The scores for phonological and phonemic awareness, lowercase rapid naming efficiency, uppercase rapid naming efficiency (posttests for both groups) and word reading efficiency (posttests for both groups) were normally distributed, but the scores for word reading accuracy, word reading efficiency (pretest for both groups), and uppercase and lowercase rapid naming were not normally distributed. Therefore, we analyzed the data using generalized mixed models—*glmer* function from lme4 package in R—due to the flexibility to sample distribution assumptions. Additionally, generalized mixed-effects model analyses take subject variability into account, which, as Bates [40] pointed out, is a consideration that is generally lacking in learning to read literature. We performed a logistic regression with test scores as fixed effects and fitted random effects for participants in our models. Additionally, we ran cross-comparisons in order to find the best adjusted model and took not only subject variability into consideration, but uncertainty distribution visualizations [41].

For the analyses of uppercase and lowercase naming (NAR) accuracy, we adjusted generalized (*poisson* family) linear mixed-effects models in order to check whether there are significant training effects on uppercase and lowercase letter naming accuracy scores. The best adjusted model for each variable was fitted in order to analyze whether GraphoGame training impacts reading development. Furthermore, we analyzed the relationship between the teacher-based dyslexia screener scores and test scores. We carried out correlation tests between the number of minutes played and training gains to test whether total time played influenced pre-posttest differences. We used Spearman and Pearson correlation according to normality distribution of the samples.

## 3. Results

The results show that word reading accuracy and lowercase rapid naming accuracy were positively and significantly impacted by GraphoGame’s practice. Moreover, students in the active control showed a decrease in their rapid naming accuracy scores for lowercase letters. 

The results show that both groups had a significant effect size in the comparison of pre and post test scores. Table 1 presents the descriptive data for the experimental group pre and posttest scores. 

The experimental group showed a strong effect size for word reading accuracy (cohen’s d = 0.81), and moderate effect sizes for uppercase naming accuracy (cohen’s d = 0.63) and phonemic awareness CONFIAS overall score (cohen’s d = 0.50). There were no other statically significant results among the remaining experimental conditions. We present Figure 2, which demonstrates the experimental group test scores among significant conditions.

We also present Table 2, which displays the descriptive data for the active control group along with their effect sizes.

A moderate effect is seen on the phonological and phonemic awareness test score (cohen’s d = 0.53). Moreover, there is a negative moderate effect for lowercase naming accuracy (cohen’s d = −0.55), which suggest participants in the control group performed worse on their posttests. We present Figure 3, which shows the active control group scores among all conditions.

### 3.1. Upper and Lowercase Letter Naming 

Our results suggest that introducing lowercase letters in GraphoGame for 1st and 2nd graders was not associated with statistically significant pre and posttest differences in lower case naming for the experimental group. However, the control group, which was not exposed to lowercase letters in class or in the app, showed a negative effect for lowercase naming. This suggests that regular instruction may have made naming lowercase letters become significantly harder for the children who were not exposed to lowercase letters. 

### 3.2. Rapid Letter Naming Accuracy and Efficiency

We found that uppercase letter naming could significantly be predicted by an interaction between moment (pre and posttests) and condition (β = 3.81735, x^2^ < 0.03). Post hoc analysis using the *emmeans* function revealed that interactions between pre and post tests for the control group were significant (*p* < 0.001, Tukey adjustment), however pre and post tests and experimental group interactions were not significant (*p* = 0.7). These results show that both groups improved their uppercase letter naming skills from pre to post test, as would be expected from the business-as-usual instruction that introduces uppercase letters only. Based on that, we cannot affirm that such gain was due to GraphoGame training, but probably due to reading instruction in the classroom in general. Figure 4 shows the pre-posttest NAR accuracy scores comparison between groups for upper and lowercase letters.

Regarding uppercase and lowercase letter naming efficiency, the best adjusted generalized mixed-effects model (gamma family) showed no interaction between condition and moment for both uppercase and lowercase letter naming reading efficiency (BETA). There was also no significant difference between pre and post scores nor experimental and control groups (BETA). Again, these results do not allow us to attribute any effect of GraphoGame training to letter naming accuracy nor efficiency, as predicted in our first hypothesis.

It is possible that the measure of naming speed, tested in seconds using a chronometer, may be too coarse to detect significant changes in naming efficiency. Letter reading was measured in seconds; however, using voice-onset instruments might have given a more fine-grained result for reading efficiency.

### 3.3. Word Reading Efficiency 

We investigated the interaction significance between moment and condition for word reading accuracy and efficiency. Figure 5 shows the scores for both groups prior and after the 6-week intervention.

The best generalized mixed-effects linear model (poisson family for accuracy and Gamma family for efficiency) shows that accuracy scores were significantly predicted by the interaction between moment and condition (β = 1.4681, x^2^ < 0.04). Further analyses suggest an increase in the slope for participant scores during the experiment. The comparison of slopes shows a significant change from pre to post testing only for the experimental group. 

Our sample distribution for word reading efficiency, however, could not be explained by the conditions nor the moments the tests were taken by the participants. Lastly, we fitted a generalized mixed-effects model for phonemic and phonological awareness accuracy (poisson family). Our results show that the data distribution cannot be explained by condition nor is there a difference between pre and post test scores (*p* = 0.9).

### 3.4. The Dyslexia Screener Correlations with Test Scores

We conducted correlation tests (Pearson and Spearman adjusted according to sample distribution) between participants’ final scores and their dyslexia screener scores. We expected the children who scored lower in the reading tests to have higher scores in their dyslexia screener, evidencing a higher risk for reading disabilities. The results show that the screener scores were significantly negatively correlated with all test scores. These results contribute to the criterion validity of the screener. Table 3 presents the results for these correlations.

The results confirm our hypothesis that the screener would show that teachers’ subjective assessment of their students reading skills would be corroborated by test scores, as seen in Figure 6. Medium negative correlation effects were found for uppercase letter naming accuracy and efficiency (r = −0.39, *p* = 0.02; r = −0.52, *p* = 0.001, respectively). Lowercase letter naming efficiency also presented a medium correlation effect (r = −0.42, *p* = 0.001). There were medium correlation effects for word reading accuracy (r = −0.53, *p* = 0.001) and efficiency (r = −0.50, *p* = 0.002). Figure 5 displays the scatterplots for these analyses.

Finally, the question whether total training time with GraphoGame impacts reading development, we documented the number of minutes each participant in the experimental group has spent playing GraphoGame and associated these minutes to reading gains. Participants in the experimental group who played a higher number of minutes appear to have gained the same as their peers who have played for a lower amount of time. We emphasize that all participants have spent the minimum number of hours playing the game at home (300 min). 

## 4. Discussion

EdTech games have increasingly become part of the classroom environment [42]. It is crucial that game developers find evidence-based methods provided by scientists in order to create their games. This study aimed to investigate whether GraphoGame, an EdTech game, helps children improve their phonemic awareness and, consequently, develop reading abilities. 

We hypothesized that children who trained with GG would present significant gains on letter recognition, word reading speed and accuracy, and phonemic and phonological awareness. Word reading accuracy seemingly improved for controls and experimental group children; however, supplementary analyses indicated that such improvement was significant only for children who played GraphoGame. This corroborates existing studies that suggest that GG helps leverage word-level decoding skills [39,40,41] and improving associations between letters and sounds [40,43]. The differences between group means were not statistically significant, however. Thus, the evidence is limited to suggesting significant within-group gains. Our findings corroborate other GraphoGame studies. Morgues and colleagues [44] found an improvement in word reading tests in the GraphoGame group that was significantly higher when compared to participants in the control group, Carvalhais and colleagues’ [32] study has also led to the same outcomes in continental Portuguese.

Students in the control group did not show an improvement in lowercase naming, but they also did not get worse. In the experimental group, however, students did get significantly worse in lowercase naming. The anecdotal premise of the practice of teaching children to read with uppercase letters only is that lowercase letters will confuse children. In the present study, that was not the case. As previously stated, in Brazil, children are taught to read using uppercase letters only until they are in 3rd grade. The experimental group did not show a significant increase in lowercase naming and efficiency, but, different from their peers in the control group, their performance did not significantly deteriorate from pre to post testing. Rather, the control group’s performance for lowercase letters did significantly decrease. Moreover, students in the active control group showed a significant decrease in lowercase naming accuracy. There was a moderate negative effect between pre and posttest scores, which may indicate that students in the active control group were more confused when compared to their peers in the experimental group. The impact of selective, rather than comprehensive instruction with upper and lowercase letters, may be negatively affecting the process of learning to read. TEACHING CHILDREN TO READ USING UPPERCASE LETTERS ONLY IS NOT TO THE ADVANTAGE OF CHILDREN (Sentence purposefully written in uppercase letters for illustration). It is clearly a practice that lacks support from actual evidence and lacks real-life context: connected text is almost never written solely using uppercase letters. 

The results also suggest that the screener, tailored and tested for screening dyslexia, may successfully predict reading difficulties. It may be used as a complementary assessment tool for policy-making decisions or, simply, to screen children’s reading performance and risk for developmental dyslexia.

School closures due to COVID10 pandemic have significantly impacted early literacy and education, in general; EdTech, in this sense, could mitigate some of the impacts during the period and, subsequently, is a promising tool for acceleration of learning [30,45,46,47]. To address the impact of school closures and the pandemic, the OECD has suggested that the learning loss has to be recovered, and one of the strategies includes implementation of an effective delivery system for remote learning has to be addressed; in this sense, to provide extra support by means of online tools, EdTech, among others [47]. The challenges brought by school closure and remote learning shifting responsibilities to families, without warning or preparation, anecdotally impacted adherence to our study. We asked parents and guardians to dedicate an extra, invaluable 15–20 min to the research. These parents were generally overwhelmed with the transfer to remote learning with elementary school children. 

We did not find a significant correlation between number of minutes played and word reading, letter recognition or phonological and phonemic awareness scores. We speculate that more fine-grained measures, such in-game accuracy and speed of progress (or, on the other hand, difficulties) could have been considered instead of number of minutes played.

## 5. Conclusions

Despite the limitations in our study, our results show the importance of researching reading development and technology with the purpose of creating scientific based learning games and software. The fact that children performed the word reading task significantly better in our study after being trained with GraphoGame corroborates the GG potential to assist the development of early literacy skills, and possibly help recover from the COVID19 school closures.

We suggest further investigation with GraphoGame as means to practice phonemic and phonological awareness considers the limitations present in this study. The first one is regarding the research design methods. Because the study was carried out during the height of the pandemic in Brazil, the loss of participants was significant. Our study started with 93 parents interested in participating in the research, however only 60 completed all pretests. Moreover, only 34 participants remained until the very last posttest and played at least 300 h throughout the experiment. Studies in-person, following the more well-known GraphoGame protocols, may preventing participant drop out. Even though we found significant effects of GG on early literacy skills, it is likely that integration with school practice, as McTigue and colleagues [27] have shown, could present better results.

We also suggest future studies use more precise measurement units, such as milliseconds for reading speed and in-game progress for adherence to the game. Hopefully, future studies of GG Brazil will adopt similar instruments, like to help for cross-study comparison.

## Figures and Tables

**Figure 1 brainsci-12-01494-f001:**
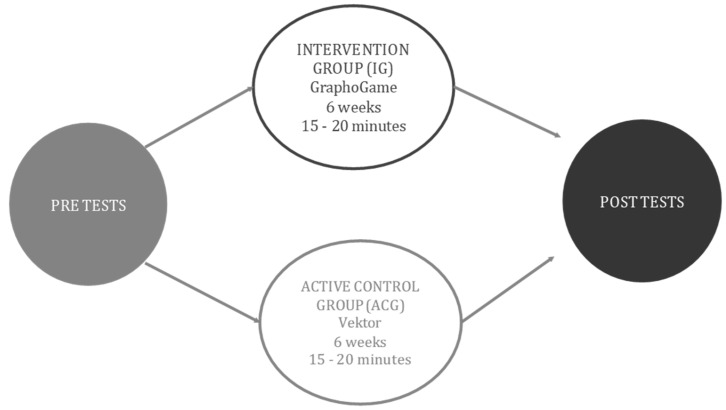
Intervention design: active control and experimental groups.

**Figure 2 brainsci-12-01494-f002:**
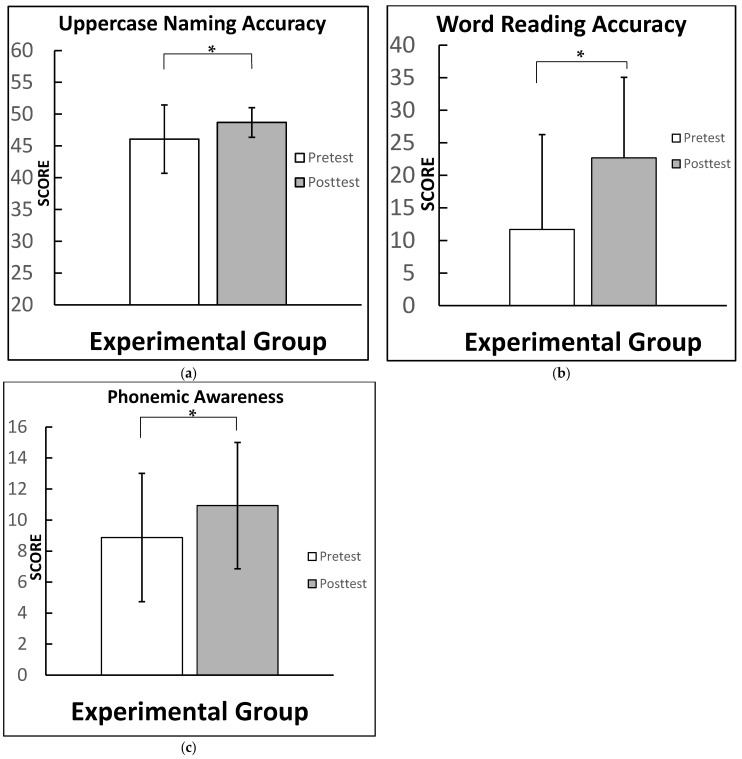
Experimental group test scores for: (**a**) uppercase naming accuracy, scores ranging from 0 to 50 (moderate effect size); (**b**) word reading accuracy, scores ranging from 0 to 36 (strong effect size) (**c**) phonemic and phonological awareness scores, scores ranging in a scale from 0 to 16 (moderate effect). Note: significant difference between test times means for the cited conditions (highlighted by *). Graphs for all conditions along with a side-by-side comparison are provided in the Appendix A (See Figure A1).

**Figure 3 brainsci-12-01494-f003:**
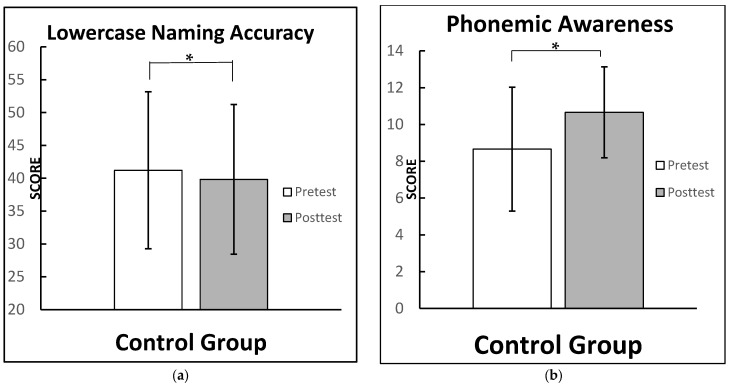
Control group test scores for: (**a**) lowercase naming accuracy, scores ranging from 0 to 50 (moderate negative effect size); (**b**) phonemic and phonological awareness scores, scores ranging from 0 to 16 (moderate effect). Note: significant difference between test times means for the cited conditions (highlighted by *). Graphs for all conditions along with a side-by-side comparison are provided in the Appendix A (See Figure A1).

**Figure 4 brainsci-12-01494-f004:**
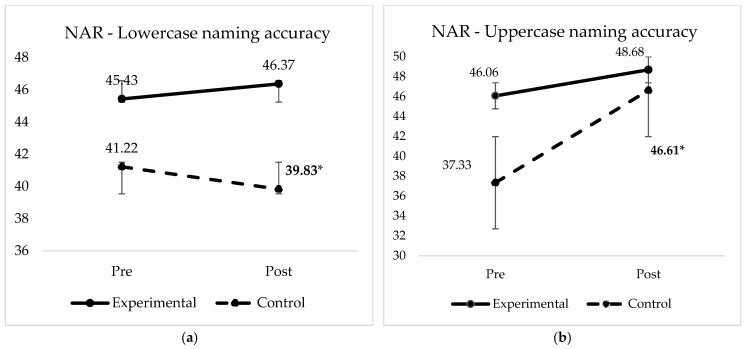
Experimental and active control groups letter naming (NAR) accuracy score mean (**a**) lowercase letter naming accuracy; (**b**) uppercase naming accuracy. Note: significant difference between group means for both uppercase and lowercase naming accuracy (* β = 3.81735, x^2^ < 0.03).

**Figure 5 brainsci-12-01494-f005:**
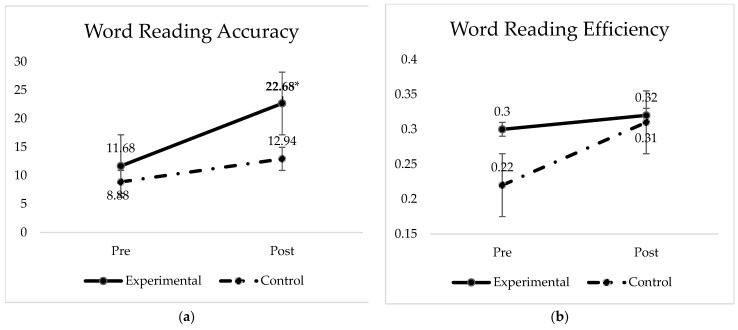
Word reading accuracy and efficiency means (**a**) TDE accuracy for both groups; (**b**) TDE efficiency for both groups. Significant training effect for word reading accuracy in the experimental group (* β = 1.4681, x^2^ < 0.04).

**Figure 6 brainsci-12-01494-f006:**
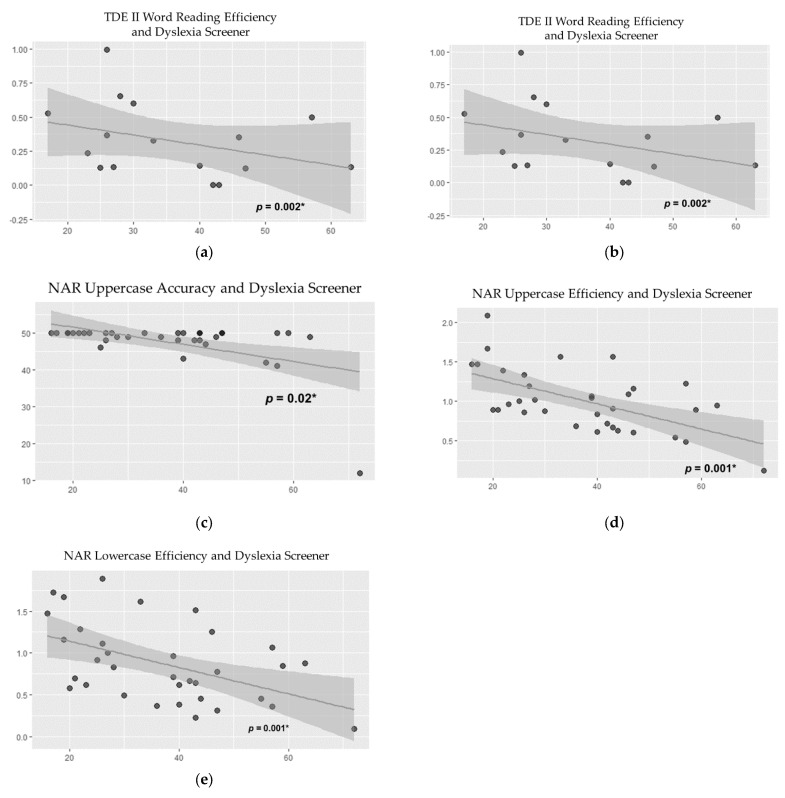
Scatterplots of the correlation between the dyslexia screener scale and the dependent variables (**a**) word reading accuracy; (**b**) word reading efficiency; (**c**) uppercase reading accuracy; (**d**) uppercase reading efficiency; (**e**) lowercase naming efficiency. Note: * = *p* < 0.05.

**Table 1 brainsci-12-01494-t001:** Experimental Group Pre and Posttest Scores and Effect Size.

Instruments and Measures	Mean Pre	Pre SD	Mean Post	Post SD	Cohen’s d ^1^
Uppercase naming accuracy	46.06	5.37	48.68	2.33	**0.63**
Uppercase naming efficiency	0.84	0.47	1.02	0.27	0.46
Lowercase naming accuracy	45.43	8.83	46.37	5.84	0.13
Lowercase naming efficiency	0.78	0.44	0.96	0.44	0.39
Word reading accuracy	11.68	14.59	22.68	12.37	**0.81**
Word reading efficiency	0.3	0.29	0.32	0.27	0.07
Phonemic awareness	8.87	4.14	10.93	4.07	**0.50**

^1^ Note: moderate to strong effects seen in uppercase naming accuracy, word reading accuracy and phonemic awareness, which are represented in bold in this table. Cohen’s d does not present moderate nor strong effects for uppercase naming efficiency (NAR), lowercase naming accuracy and efficiency (NAR) and word reading efficiency (TED II).

**Table 2 brainsci-12-01494-t002:** Instruments and measures descriptive data for active control group.

Instruments and Measures	Mean Pre	Pre SD	Mean Post	Post SD	Cohen’s d ^1^
Uppercase naming accuracy	37.33	15.23	46.61	8.97	0.07
Uppercase naming efficiency	0.63	0.56	1.00	0.48	0.32
Lowercase naming accuracy	41.22	11.95	39.83	11.40	**−0.55**
Lowercase naming efficiency	0.65	0.47	0.79	0.47	0.01
Word reading accuracy	8.88	13.33	12.94	13.63	0.09
Word reading efficiency	0.22	0.27	0.31	0.30	0.04
Phonemic Awareness	8.66	3.37	10.66	2.47	**0.53**

^1^ Note: Positive moderate effect for phonemic awareness and negative moderate effect for lowercase naming accuracy, which are here highlighted in bold. Cohen’s d does not present moderate nor strong effects for uppercase naming accuracy or efficiency (NAR), lowercase naming efficiency (NAR) and word reading accuracy or efficiency (TED II).

**Table 3 brainsci-12-01494-t003:** Correlation between the dyslexia screener and the dependent variables.

Instruments and Measures	ELE Scale ^1^	*p* Value
Uppercase letter naming accuracy	**−0.39**	**0.02**
Uppercase letter naming reading efficiency	**−0.52**	**0.001**
Lowercase letter naming accuracy	−0.27	0.1
Lowercase letter naming reading efficiency	**−0.49**	**0.002**
Word reading accuracy	**−0.53**	**0.001**
Word reading efficiency	**−0.50**	**0.002**
CONFIAS	−0.29	0.08

^1^ Dyslexia screener (ELE) score measures frequency, where “always” was attributed 5 points and “never” 1 point. Thus, lower scores are related to better reading and writing skills. Medium correlation effects for uppercase letter naming accuracy and efficiency, lowercase letter naming efficiency, and word reading accuracy and efficiency were highlighted in bold.

## Data Availability

The data presented in this study are available on request from the corresponding author. The data are not publicly available due to ethical reasons.

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
