# Peer review of "Well Played! Promoting Phonemic Awareness Training Using EdTech—GraphoGame Brazil—During the COVID-19 Pandemic"

_brainsci, 2022, doi:10.3390/brainsci12111494_

Round 1

Reviewer 1 Report

I find the paper interesting to read.  The use of the GraphoGame program is interesting and the application to Portuguese-speaking Brazilian children is informative.  The article is clearly written although editing from an English-speaking person would be appropriate.  The description of the results should be improved.  In particular, the authors carry out complex statistical analyses (linear mixed-effects models analyses) which are appropriate for their data.  However, they do not fully report these analyses and it is not clear when the group differences in gain from the training are indeed significant or not.  I would ask the authors to clarify this part of their paper.  Additional, specific comments are listed below.

Line 207 Participants either withdrew during (incomplete pretests),

I think it should be Participants WHO either withdrew during (incomplete pretests),

372 The experimental group showed a strong effect size for word reading accuracy (Cohen’s d = 0,81), …

Decimal points should be marked with a “.” not with a “,”.  I would ask the authors to check this throughout their paper.

371. The results show that both groups had a significant effect size in the comparison of pre and post-test scores. Actually, figure 2 only shows the data of the experimental group.  Given that “both groups had a significant effect size” one would expect to find the performance of both groups in Figure 2.

Some of the data of the control group are presented in what is probably Figure 3 (but note that the figure is not numbered in the text and has no legend).  The problem I see is that Figure 3 shows different parameters than Figure 2.  In this way, the reader does not have the possibility to compare the results on “NAR Uppercase Naming Accuracy” in the experimental and control group (the same applies to the other categories).

The authors may enlarge figure 2 including the control group’s data or may enlarge figure 3 by including all the categories shown in figure 2 (whether or not they show relevant changes).

Table 1 shows the data of the experimental group and table 2 those of the control group.

Note that the two tables do not present the same categories.  Please add the missing ones.

Results lines 364-392

The authors present the separate Cohen d’s size effects for the two groups but do not provide statistics as to whether the group differences are actually significant.  It would seem important to specify the group changes which were significant and those in which there was a significant group difference in performance gains.  Direct comparisons of d changes without corresponding significant values do not fully express the present results.

Line 447 poison should be “poisson”.

In figure 6 one sees the changes in TDE accuracy and efficiency, respectively.  The corresponding text clarifies that the changes in accuracy are significant while those in efficiency are not. 

This difference is not visually apparent in the figures.  Presumably, this is because variability is different in the two measures.  I would suggest adding variability measures (SD or confidence intervals) to Figure 6.

Discussion

In the abstract, the authors state: "The present study aimed to investigate the impact of GraphoGame Brazil on fostering early literacy skills during the height of COVID19 school closures, in Brazil.“

Is there any difference the authors believe there was in their study because of the pandemic? Was Covid 19 just an accidental co-occurrence or can they make any more specific comments on this question? I understand it may be difficult to provide a direct comparison between a study done in a Covid and in a non-Covid period.  However, possibly the authors may be able to provide some ideas as to whether training was hindered by the pandemic or maybe even the opposite, i.e., the children found it interesting to do “something” while they were at home.  At any rate, as Covid 19 is even in the title of the paper, one would expect some more comments on its impact, or on possible other related considerations.

In the limitations of their study, the authors mention that “Because the study was carried out during the height of the pandemic in Brazil, the loss of participants was significant.”  But are they sure that this is the case?  Is there any evidence that adherence to online training is usually higher in Brazil in the non-Covid period?  I suspect that given the relatively high level of the minimum number of hours, the obtained loss may actually be expected.  At any rate, I simply ask the authors to develop more these arguments (possibly not only in the limitations section).

Author Response

We have attached an item-by-item reply.

Reviewer 2 Report

Thank you very much for the opportunity to review such a current, pertinent and interesting paper.

With a clear introduction and theoretical framework, the paper would gain in quality if a more up-to-date bibliography were presented, which would also allow for a more in-depth analysis of the data. Of the 43 references presented, only 27% refer to the last 5 years. Thus, it is suggested:

- update the bibliography, including more recent studies in the area;

- deepen the discussion, in the light of these recent studies, and considering the later and lasting impact of the use of the EdTech - GraphoGame, the role of the guardians in the process (can the children's evolution not be explained by the parents' participation in other activities that promote phonemic awareness?), and the authors' reasoned position regarding the number of hours children should spend in front of the screen for the EdTech - GraphoGame to be really effective.

It is also suggested that the data from the graphs for the experimental and control groups be integrated into single graphs that allow the differences to be seen more clearly and directly.

Author Response

We have attached an item-by-item reply.
